# The Effect of Tactile Imagery Training on Reaction Time in Healthy Participants

**DOI:** 10.3390/brainsci13020321

**Published:** 2023-02-14

**Authors:** Kishor Lakshminarayanan, Vadivelan Ramu, Janaane Rajendran, Kamala Prasanna Chandrasekaran, Rakshit Shah, Sohail R. Daulat, Viashen Moodley, Deepa Madathil

**Affiliations:** 1Neuro-Rehabilitation Lab, Department of Sensors and Biomedical Engineering, School of Electronics Engineering, Vellore Institute of Technology, Vellore 632014, India; 2Department of Chemical and Biomedical Engineering, Cleveland State University, Cleveland, OH 44115, USA; 3University of Arizona College of Medicine, Tucson, AZ 85724, USA; 4Arizona Center for Hand to Shoulder Surgery, Phoenix, AZ 85004, USA; 5Jindal Institute of Behavioural Sciences, O. P. Jindal Global University, Haryana 131001, India

**Keywords:** reaction time, motor imagery, tactile imagery, vibration, training

## Abstract

Background: Reaction time is an important measure of sensorimotor performance and coordination and has been shown to improve with training. Various training methods have been employed in the past to improve reaction time. Tactile imagery (TI) is a method of mentally simulating a tactile sensation and has been used in brain–computer interface applications. However, it is yet unknown whether TI can have a learning effect and improve reaction time. Objective: The purpose of this study was to investigate the effect of TI on reaction time in healthy participants. Methods: We examined the reaction time to vibratory stimuli before and after a TI training session in an experimental group and compared the change in reaction time post-training with pre-training in the experimental group as well as the reaction time in a control group. A follow-up evaluation of reaction time was also conducted. Results: The results showed that TI training significantly improved reaction time after TI compared with before TI by approximately 25% (pre-TI right-hand mean ± SD: 456.62 ± 124.26 ms, pre-TI left-hand mean ± SD: 448.82 ± 124.50 ms, post-TI right-hand mean ± SD: 340.32 ± 65.59 ms, post-TI left-hand mean ± SD: 335.52 ± 59.01 ms). Furthermore, post-training reaction time showed significant reduction compared with the control group and the improved reaction time had a lasting effect even after four weeks post-training. Conclusion: These findings indicate that TI training may serve as an alternate imagery strategy for improving reaction time without the need for physical practice.

## 1. Introduction

Reaction time is the total length of time taken by an individual to observe an external stimulus, decode that information, make an appropriate decision, and initiate a motor activity as a response to the stimulus [1]. Reaction time plays a crucial role in the day-to-day, real-time activities of professionals such as soldiers, security guards, drivers, and pilots in terms of performance or response. In neurophysiology, reaction time is a reliable measure of an individual’s sensorimotor performance and coordination [2]. There is a strong need to reduce reaction times in the general population, especially in athletes such as swimmers, sprinters, and volleyball players. Also of importance are patients with diabetes or osteoporosis, to prevent falls [3], people with intellectual disabilities, and people with hearing loss or visual impairments [4,5,6]. With its importance evident in performance enhancement and rehabilitation, various training methods have been employed in the past to improve reaction time, such as training inside virtual reality environments [7], step training where elderly participants are instructed to perform a single or multiple voluntary steps in an upright position in response to a stimulus [8], balance and mobility training [9], and working memory training where the participants are instructed to observe, recall, and respond to visual cues [10]. Recently, motor imagery (MI) training, where participants are made to imagine a movement or activity without a concomitant physical movement, Guillot et al. [11] has shown a lot of promise in improving reaction time when combined with action observation [5,6], physical practice [12], and MI-based brain computer interface (BCI) training [13].

Studies have shown that tactile stimuli can improve performance of various tasks [14,15] including reaction time in stimulus detection reaction time paradigms [16,17]. Such tactile cues not only showed an improvement in simple reaction time tasks [18], but also in changing an on-going reach more rapidly than visual cues [19]. However, there are individuals who have lost their sense of touch and proprioception and rely solely on visual cues for feedback. This leads to slow and inaccurate movements which require significant concentration to execute correctly [20]. In such instances, to see a faster reaction time than from the visual cues, the tactile stimulation needs to be applied to the body part not affected by the sensory impairment, which might not fit every circumstance and might even prove to be cumbersome. An alternative approach would be to imagine the tactile stimulation similar to imagining a motor activity in MI training.

Imagining a tactile sensation has been reported to elicit identical cortical responses to MI [21] and have been successfully employed as a new strategy for BCI applications [22]. This tactile imagery (TI) recruits the primary somatosensory cortex [23], while MI recruits the motor cortex. While MI mentally simulates physical movement through imagery of a body part, TI maintains somatosensory attention to a particular body part. Therefore, TI might be a novel imagery strategy for individuals with reduced sensory abilities. However, it is yet unknown whether TI can improve reaction time, the quantification of which would contribute to the development of a novel paradigm to reduce reaction time.

Therefore, the purpose of the current study was to investigate the effect of TI as a training regimen in improving the reaction time in healthy participants. To achieve this, we examined the reaction time to vibratory stimuli before and after a TI session in an experimental group and compared the change in reaction time post-TI with pre-TI in the experimental group as well as the reaction time in a control group. Furthermore, a follow-up evaluation of reaction time was also conducted on the experimental group to examine the long-lasting effects of TI. It was hypothesized that TI will significantly reduce reaction time and the learning effects will be present after several weeks post-TI.

## 2. Methods

### 2.1. Participants

A total of 70 healthy adults read and signed a written informed consent form before participating in the experiment. The protocol was approved by the Vellore Institute of Technology Review Board. All subjects verbally disclosed that they had no history of upper limb injury or musculoskeletal or neurologic disorders. The subjects had no prior experience with tactile imagery. The subjects were randomly assigned to two groups: (i) experimental group (23 males and 12 females, mean ± SD age of 24.43 ± 6.92 years) and (ii) control group (24 males and 11 females, mean ± SD age of 21.80 ± 6.90 years).

### 2.2. Procedure

Reaction times to a vibrotactile stimulus to either the left- or right-hand index finger pads were measured before and after a tactile imagery session in the experimental group and compared with the reaction time of the control group.

### 2.3. Equipment and Set-Up

Two buttons were fixed on a horizontal rod (length: 50 cm, diameter: 5 cm) on either side of the center of the horizontal rod at a distance of 6 cm from each other. This response device was fixed on a support (length: 50 cm, width: 22 cm, height: 10 cm) and placed on a table (Figure 1). Tactile stimulation was delivered via two flat vibration micro motors (Sunrobotics, Ahmedabad, India) that were fastened using skin-safe tape to the left- and right-hand index finger pads. The vibration motors were controlled using an Arduino UNO R3 microcontroller that sent a short 5 V square pulse lasting 100 ms to the vibration motors to deliver a short vibrotactile stimulus to the index finger pads. At the same time, the Arduino microcontroller also sent a square pulse of the same amplitude and duration to a data acquisition (DAQ) board (USB DAQ 6210, National Instruments, Austin, TX, USA) to record the times at which the vibrotactile stimuli was delivered to the finger pads. Similarly, the buttons were also connected to the DAQ board and produced a square pulse of 3 V that lasted as long as the buttons were pushed down. The DAQ board received the square pulses from the microcontroller corresponding to the two vibration motors and the two buttons in four separate channels. The data were displayed using a LabVIEW code (National Instruments, Austin, TX, USA) that recorded the data at a frequency of 1000 Hz.

### 2.4. Experimental Design

Subjects were made to sit comfortably at the table with their hands parallel and palms down on the response device, so that the thumbs of each hand could press on a button. The experiment was divided into three sessions, namely pre-TI reaction time (RT-pre), tactile imagery training (TI), and post-TI reaction time (RT-post). In the pre-TI and post-TI sessions, subjects were presented with a short vibrotactile stimulus to either the left- or right-hand index finger pads randomly. Subjects were instructed to react as quickly as possible to the stimulus by pressing on the corresponding left or right button. The next stimulus was presented after an interval of 3 s. There were five blocks in each session with each block having ten consecutive trials with five trials each for the left and right hands presented randomly. In the TI-training session, subjects were instructed to imagine a tactile sensation to either the left- or right-hand index finger pad. The session had ten blocks of fifteen consecutive trials, with five blocks each for the left and right hands. In each block, a short vibrotactile stimulus lasting 100 ms was presented to either the left or right index finger pad for the first five trials, following which subjects were instructed to imagine the stimulus for the remaining ten trials at the same finger pad the stimulus was presented to. The trial duration was the same as the pre-TI and post-TI sessions, with a short auditory cue indicating the start of each trial to the subject. Adequate rest was provided between each block in all the sessions to avoid fatigue.

The current study tried to evaluate the training effect TI has on reaction time in healthy individuals. However, the repeated trials in the pre-TI and post-TI sessions to evaluate the reaction time where subjects were asked to press a button as soon as they perceived a tactile stimulation might have a training effect as well. Hence, the control group performed the same number of trials with the tactile stimulation as the experimental group to account for the learning effect from the tactile stimulation-based reaction time sessions (pre-TI and post-TI). Subjects in the experimental group performed an additional TI session along with the pre-TI and post-TI sessions, while subjects in the control group only performed two reaction time sessions similar to pre- and post-TI with five blocks each (RT-c1 and RT-c2) and a total of ten blocks with a break between the sessions. As such, the training effect of the TI sessions will be seen in the reaction times from the post-TI session. Therefore, for the comparison between the experimental and the control groups, the average reaction time from the first five blocks in the control group (RT-c1) were considered to correspond to the average reaction time from the five blocks in the pre-TI session (RT-pre) in the experimental group and the last five blocks (RT-c2) were considered to correspond to the post-TI session (RT-post).

Subjects who participated in the experimental group answered a modified version of the Vividness of Movement Imagery Questionnaire (VMIQ) [24] that was altered to include five items relevant to tactile imagery instead of motor imagery. The items are as follows:Item 1 evaluated the vividness of tactile imagery in the right-hand index finger.Item 2 evaluated the same as item 1 but for the left-hand index finger.Item 3 evaluated the consistency with which the subjects were able to imagine and maintain the tactile imagery following the vibratory stimulus at the beginning of each block.Item 4 evaluated any improvement in reaction time the subjects felt during the post-TI session.Item 5 evaluated which hand the subjects were able to imagine the tactile imagery better with.

The subjects were instructed to answer all the items without conferring with others. The questionnaire used the five-point scale developed by Marks et al. [25] to assess the imagery vividness.

The reaction time and VMIQ assessments were all performed on the same day as a single experiment. Furthermore, to study the lasting effects of the TI training, subjects in the experimental group were called back for a follow-up session four weeks after the initial experiment. However, there were ten drop-offs and 25 out of the 35 subjects showed up for the follow-up evaluation. The reaction times of the experimental group subjects were evaluated again in a session (RT-e) similar to the control group where the subjects performed five blocks of reaction time trials.

### 2.5. Data Analysis

The data acquired by the DAQ board was exported as an excel sheet. Reaction time was calculated as the time between the start of the rising edge of the vibration square pulse delivered to either the left- or right-hand index finger pad and the button push-generated square pulse from the corresponding left- or right-hand thumb. The trials in which the subjects failed to press the button or incorrectly pressed the wrong side button were removed. Finally, the reaction times were averaged for all the remaining trials in each reaction time session, namely RT-pre, RT-post, RT-c1, RT-c2, and RT-e for each hand.

For statistical analysis, paired *t*-tests were performed to compare the difference in reaction times within the experimental group between the pre- and post-TI training (RT-pre vs. RT-post) and within the control group (RT-c1 vs. RT-c2). The reaction times between the initial experiment and the follow-up evaluation (RT-pre vs. RT-post, RT-pre vs. RT-e, and RT-post vs. RT-e) were compared using paired *t*-tests only for the 25 subjects from the experimental group who returned for the follow up. Furthermore, two sample *t*-tests were performed to compare between the experimental and control groups (RT-pre vs. RT-c1 and RT-post vs. RT-c2). The *t*-tests were performed for the right and left side individually. The statistical analysis was performed using SigmaStat 4.0 (Systat Software Inc., San Jose, CA, USA). An α level of 0.05 was considered for statistical significance.

## 3. Results

As a preliminary step, the validity of the assumptions made regarding the reaction time results was verified. No significant outliers were detected. The Shapiro–Wilk normality test confirmed that the normality assumption was not violated for all groups (*p* > 0.05). The condition of equal variances between different experimental conditions is known as sphericity. A Brown–Forsythe test was conducted to test for this assumption and confirmed equal variance (*p* > 0.05).

The results of the statistical analysis indicate that the reaction time from RT-post (right hand mean ± SD: 340.32 ± 65.59 ms, left hand mean ± SD: 335.52 ± 59.01 ms), which represents the reaction time of the experimental group after the TI training, was significantly reduced for both hands (*p* < 0.01) compared with RT-pre (right hand mean ± SD: 456.62 ± 124.26 ms, left hand mean ± SD: 448.82 ± 124.50 ms), which represents the reaction time of the experimental group before the TI training (Figure 2A). Similarly, in the control group RT-c2 (right hand mean ± SD: 381.72 ± 78.01 ms, left hand mean ± SD: 374.64 ± 77.07 ms) was significantly smaller (*p* < 0.01) than RT-c1 (right hand mean ± SD: 425. ± 87.76 ms, left hand mean ± SD: 417.02 ± 78.26 ms) for both hands (Figure 2B). Between the experimental and control groups, while there was no statistical difference (*p* = 0.22 for right hand and *p* = 0.20 for left hand) in the average reaction times between RT-pre and RT-c1 (Figure 3A), the average reaction time in the experimental group post-TI (RT-post) was significantly smaller (*p* < 0.01) than RT-c2 for both hands (Figure 3B).

Furthermore, the paired *t*-tests performed to compare the difference in reaction times between the pre-TI, post-TI training, and the follow-up evaluation (RT-pre vs. RT-post, RT-pre vs. RT-e, and RT-post vs. RT-e) showed that there was a significant reduction in reaction time from RT-post (*p* < 0.01) and RT-e (*p* < 0.01) compared with RT-pre and no statistically significant difference between RT-post and RT-e (right hand mean ± SD: 339.01 ± 57.72 ms, left hand mean ± SD: 335.29 ± 51.91 ms) for both the right (*p* = 0.66) and left (*p* = 0.39) hands for the experimental group (Figure 4).

The results of the modified VMIQ administered to the experimental group showed that all subjects rated the tactile imagery favorably. On item 1, which evaluated the vividness of tactile imagery in the right hand, the mean score was found to be 3.74 ± 1.03 out of 5, indicating that the majority of subjects found the tactile imagery in their right hand to be very vivid. Similarly, on item 2, which evaluated the vividness of tactile imagery in the left hand, the mean score was also found to be 3.68 ± 0.93 out of 5. On item 3, which evaluated the consistency with which the subjects were able to imagine and maintain the tactile imagery following the vibratory stimulus at the beginning of each block, the mean score was 3.51 ± 0.78 out of 5, indicating that the majority of subjects found the tactile imagery to be consistent and easy to maintain throughout the course of the experiment. On item 4, which evaluated any improvement in reaction time the subjects felt during the post-TI session, the mean score was 3.62 ± 1.11 out of 5, indicating that the majority of subjects felt an improvement in their reaction time following the TI training. Finally, on item 5, which evaluated which hand the subjects were able to imagine the tactile imagery better with, 20 out of the 35 subjects reported that they were able to imagine the tactile imagery well with their right hand and 13 reported their left hand, while 2 reported none. The Spearman correlation coefficients between items 1-4 and the respective percentage difference in reaction time between the pre-TI and post-TI for the right and left hands are shown in Table 1. The responses to the items were positively and highly correlated with reaction time percentage difference. Item 5 which evaluated the hand subjects felt they were able to imagine the tactile imagery better showed no correlation with the reaction times and the handedness of the subjects (Rho = 0.130, *p* = 0.455).

## 4. Discussion

The current study aimed to investigate the effect of tactile imagery (TI) as a training regimen on reaction time in healthy participants. The results showed that the experimental group, who underwent a TI training session, had a significant decrease in reaction time to vibratory stimuli compared with both their pre-training reaction time and the reaction time of the control group. Furthermore, the study also included a follow-up evaluation of reaction time four weeks after the training session, which showed that the reduction in reaction time seen in the experimental group persisted over time. These findings suggest that TI training may be an effective strategy for improving reaction time in healthy individuals and that the effect of TI training on reaction time may have a lasting effect.

The reaction time is a complex neuromotor skill that can be influenced by a variety of external and internal factors such as the type of stimulus (auditory, visual, or tactile), sex, age, physical fitness, fatigue level, distraction, alcohol, personality type, dominant hand, biological rhythm, and health [26]. A correlation between training regimes and reaction times has been shown in several studies. Students who frequently exercised were shown to have shorter reaction times than those who maintained sedentary lifestyles [27]. When compared with pre-training, older diabetes individuals with or without neuropathy who participated in a moderate or intense supervised exercise program showed shorter simple reaction times [28]. Similar to this, children and adolescents with modest intellectual disabilities who took part in physical fitness training programs showed improved reaction times when compared with a control group [29]. In the current study, the repeated trials from the reaction time tests has shown to have a training effect on both the experimental and control groups with both groups showing a reduction in reaction time at the end of the study. However, the addition of TI had a significant training effect on the reaction time of the experimental group. The current study adds to the growing body of literature on the use of motor imagery training and other mental imagery techniques for improving various motor abilities, including reaction time [5,6,13,30,31]. The current study is unique in that it specifically examines the use of tactile imagery as a novel imagery strategy to improve reaction time. The results suggest that TI may be an effective strategy for individuals with reduced sensory abilities. This is particularly important for populations such as stroke survivors, who have been reported to show reduced ability to imagine movement in their affected hand [32].

Furthermore, the lasting effect of TI training on the reaction time seen in our study is consistent with previous research on the lasting effects of MI training on reaction time [5,6,10,33]. A study by Kraeutner et al. [33] found that MI training resulted in improved reaction time, and the improvement persisted for at least four weeks after the training. Similarly, a study by Xiu et al. [10] found that working memory training, which included elements of MI, resulted in improved reaction time that persisted for one month after the training. The lasting effect of TI training on reaction time may be related to the changes in neural activity that occur as a result of the training. Research has shown that MI recruits the primary motor and premotor cortices [33], while TI recruits the somatosensory cortex [23]. Such recruitment might lead to changes in neural activity that are believed to underlie the lasting effect of imagery training. The results from the current study suggest that TI training, like MI, may also lead to similar changes in neural activity and these changes may also contribute to the lasting effect of the training on reaction time.

In addition to measuring reaction time, the current study also included a modified vividness test on TI, where the experimental group rated their ability to imagine the tactile sensation on a scale of 1–5, with 5 being highly vivid. The results of the vividness test showed that the experimental group rated their ability to imagine the tactile sensation highly for both right and left hands. The scores for items 1–4 also showed high and positive correlation with the reaction time. Moreover, the majority of the subjects felt an improvement in the reaction time in both hands when reacting to the vibratory stimulus during post-TI. Interestingly, although the majority of subjects reported that they felt improvement in reaction time more in their right hand compared to with their left, the score did not correlate with either the reaction times or the subjects’ handedness, suggesting that the improvement was seen in both hands similarly in spite of what the subjects felt. The results from the questionnaire suggest that the participants in the experimental group were able to effectively engage in TI during the training session. The ability to effectively engage in imagery is an important factor in determining the effectiveness of the imagery training. Research has shown that the vividness of imagery is positively correlated with the magnitude of neural activity in the somatosensory cortex [34] and with the effectiveness of the imagery training in improving motor abilities [30]. The high vividness scores in the experimental group in this study suggests that it may have contributed to the improvement in reaction time seen in the experimental group. Furthermore, the high vividness score in the experimental group may also indicate that the TI training was well designed and effectively implemented, which is important for the generalization of the study findings. The subjects in the experimental group rating their ability to imagine the tactile sensation highly suggests that they were able to effectively engage in the TI training and that the training was successful in increasing their ability to imagine the tactile sensation.

The current study has a few limitations, namely the study only examined the effect of TI on reaction time in healthy participants. Future studies are needed to investigate the effectiveness of TI in other populations, such as people with diabetes or osteoporosis, people with intellectual disabilities, and people with hearing loss or visual impairments and for other motor disabilities. Additionally, the study only used vibratory stimuli as the external stimulus, and future research could examine the effect on reaction time using other types of stimuli, such as visual or auditory stimuli.

## 5. Conclusions

In conclusion, the current study provides evidence that TI training can improve reaction time and that the effect of the training persists over time. The high vividness scores in the experimental group in this study suggests that the participants were able to effectively engage in the TI training, that the training was successful in increasing their ability to imagine the tactile sensation, and that this may have contributed to the improvement in reaction time seen in the experimental group. This suggests that TI may be a promising strategy for improving reaction time in healthy individuals and the findings of this study may have important implications for improving reaction time in both healthy and patient populations needing improvement in their reaction time. However, more research is needed to understand the neural mechanisms underlying the effect of TI training on reaction time and to investigate the effectiveness of TI training in other populations and for other motor abilities.

## Figures and Tables

**Figure 1 brainsci-13-00321-f001:**
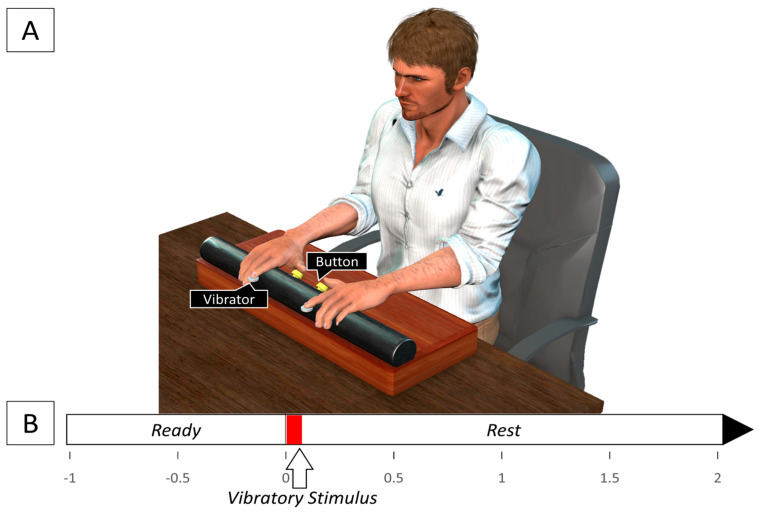
(**A**) Experimental set-up; (**B**) Timeline of events for a single trial. The vibratory stimulus (indicated by the red area) is presented for 100 ms at the beginning of each trial randomly to either the right or left index finger pad.

**Figure 2 brainsci-13-00321-f002:**
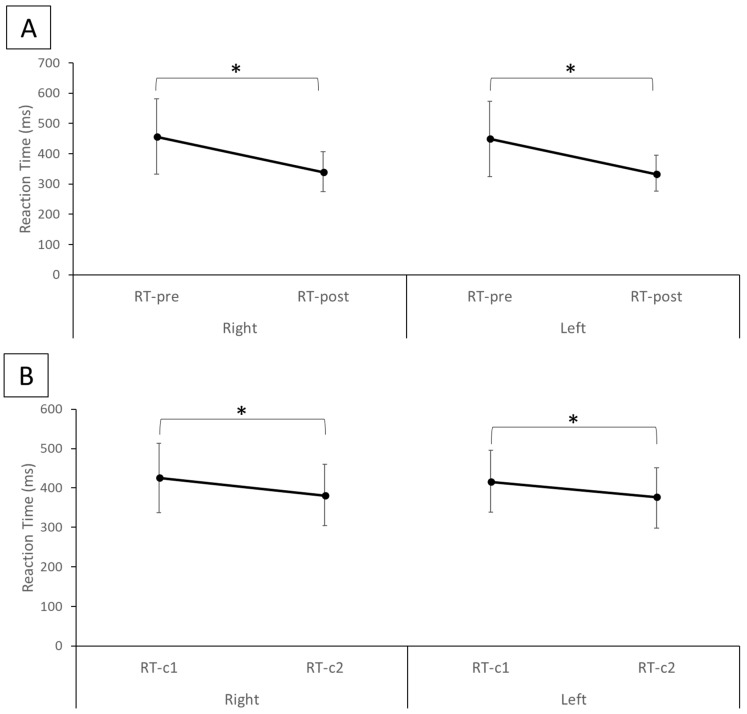
Mean ± SD reaction times for (**A**) the experimental group before (RT-pre) and after (RT-post) tactile imagery training and (**B**) the control group first five blocks (RT-c1) and last five blocks (RT-c2) of reaction time evaluation for the right and left hand. * Statistical significance with *p* < 0.05.

**Figure 3 brainsci-13-00321-f003:**
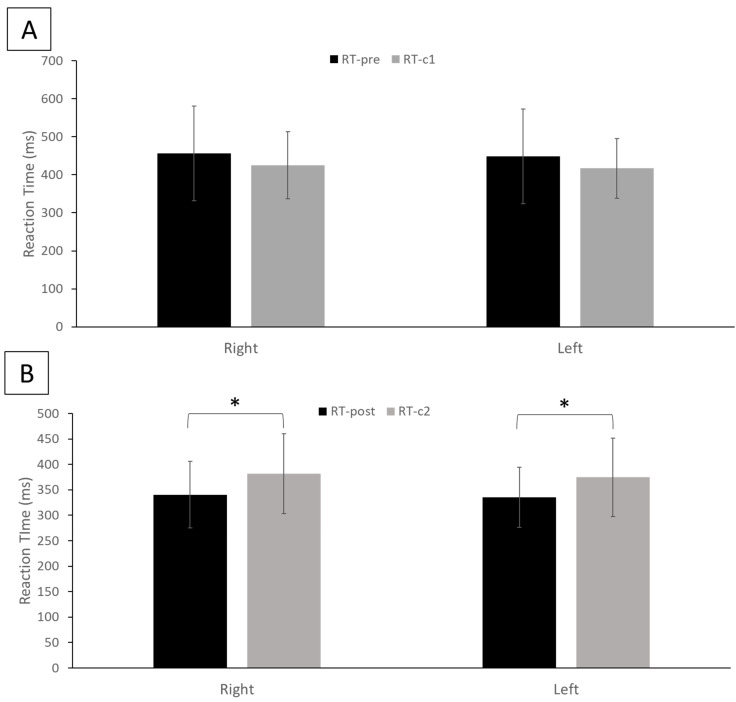
Comparison between (**A**) the mean ± SD reaction time from the experiment group (RT-pre) and the control group (RT-c1) for each hand and (**B**) the mean ± SD reaction time from the experiment group (RT-post) and the control group (RT-c2) for each hand. * Statistical significance with *p* < 0.05.

**Figure 4 brainsci-13-00321-f004:**
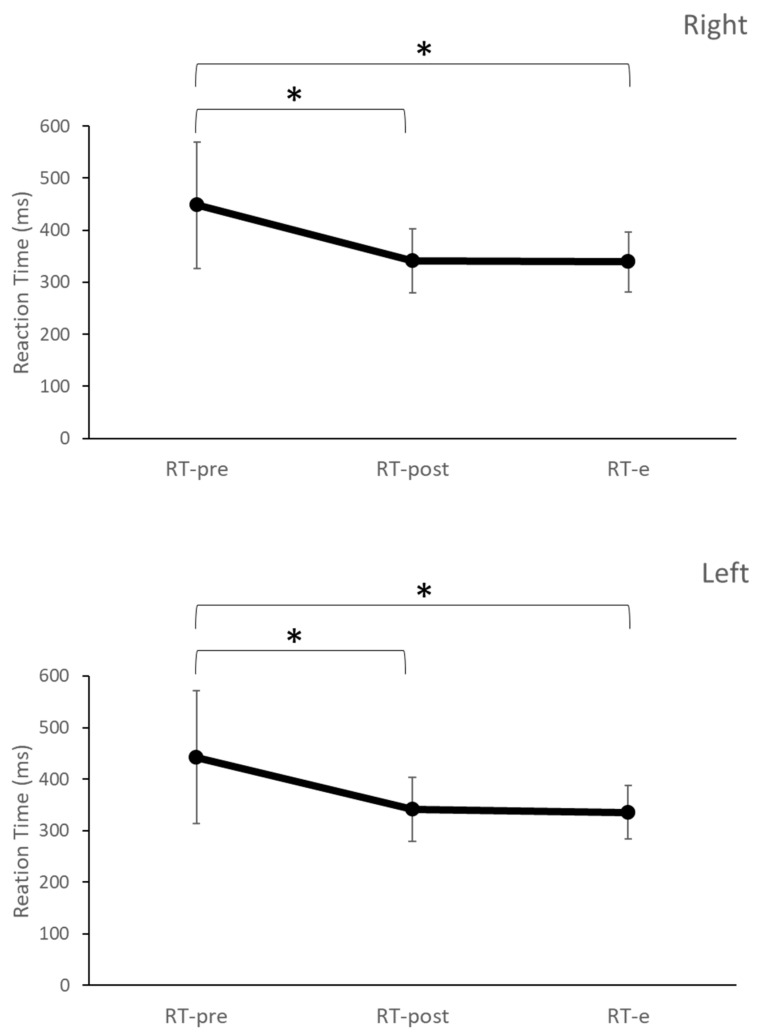
Mean ± SD reaction time from the 25 subjects from the experimental group during before TI training (RT-pre), after TI training (RT-post), and follow-up evaluation (RT-e) for each hand. * Statistical significance with *p* < 0.05.

**Table 1 brainsci-13-00321-t001:** Spearman’s Rho correlation coefficients computed between the items in the VMIQ and the right- and left-hand reaction time percentage difference.

	Percentage Reaction Time	
	Right-Hand	Left-Hand
Item 1	0.641 (*p* = 0.00003)	-
Item 2	-	0.483 (*p* = 0.003)
Item 3	0.671 (*p* = 0.000006)	0.487 (*p* = 0.003)
Item 4	0.956 (*p* = 0.0000002)	0.728 (*p* = 0.0000002)
Item 5	0.0972 (*p* = 0.576)	0.0743 (*p* = 0.670)

## Data Availability

The datasets used and/or analyzed during the current study are available from the corresponding author on reasonable request.

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
