# Peer review of "The Effect of Tactile Imagery Training on Reaction Time in Healthy Participants"

_brainsci, 2023, doi:10.3390/brainsci13020321_

Round 1

Reviewer 1 Report

Lakshminarayanan et al. report a new study focused on the effect of training on reaction time in healthy participants. In general, this manuscript needs more work prior to consideration for publication.    Here are some of my comments:   Abstract: There are no specifics provided here. What was the reaction time measured in? How much did it differ? Mean and Std. Deviation? What % change was observed?   Introduction: Same comment as above. Details of previous studies are missing. Clearly, there are a number of studies in literature based on tactile imaging and reaction time. Please include them here. What is the knowledge gap and motivation for this study? What is unknown from tactile stimulation literature? Please elaborate.    Methods: Were the trials randomly performed? If so, how did it vary for the participants? Did the participants ever receive a sham trial (a negative impulse between the stimulus)?    Data analysis: Was the normality check performed? The authors should test for normality and use the appropriate statistical tests. Kindly rectify.    Results: All box plots shown here should be changed to Bee Swarm Box Plot (i.e. showing the actual data points).    Figure 5 is worthless (it's a multi-panel figure without any sub labels) unless some important message is being conveyed by the authors. Revise and present the data in an acceptable manner that would improve the quality of the manuscript or else provide a table.   

Author Response

We thank the reviewers for providing thoughtful comments. We have responded to the comments point-by-point. The corresponding changes have been made in the revised manuscript.

Abstract: There are no specifics provided here. What was the reaction time measured in? How much did it differ? Mean and Std. Deviation? What % change was observed?   

The details have been updated and the abstract has been revised.

Introduction: Same comment as above. Details of previous studies are missing. Clearly, there are a number of studies in literature based on tactile imaging and reaction time. Please include them here. What is the knowledge gap and motivation for this study? What is unknown from tactile stimulation literature? Please elaborate.    

We have revised the introduction to include more literature on tactile imagery and reaction time studies using tactile stimulation. Furthermore, we have reworked the knowledge gap as the lack of information on the training effect of tactile imagery on reaction time and have updated the motivation accordingly.  

Methods: Were the trials randomly performed? If so, how did it vary for the participants? Did the participants ever receive a sham trial (a negative impulse between the stimulus)?    

The trials were performed randomly within each block of reaction time assessment. The left or right index fingers were randomly stimulated within a block and the subjects had to react to the stimulation and press the respective left or right button. We did not include a sham trial since the trials were for evaluating the reaction time speed.  

Data analysis: Was the normality check performed? The authors should test for normality and use the appropriate statistical tests. Kindly rectify.    

The normality and variance check were performed and the data was found to be normal and of equal variance. The information has been updated under Results in lines 195-199.

Results: All box plots shown here should be changed to Bee Swarm Box Plot (i.e. showing the actual data points).    Figure 5 is worthless (it's a multi-panel figure without any sub labels) unless some important message is being conveyed by the authors. Revise and present the data in an acceptable manner that would improve the quality of the manuscript or else provide a table.   

We can understand how a bee swarm plot will offer information about individual data points. However, we feel that the bar graph provides a better bigger picture. We would like to appeal to the reviewer to reconsider but we will gladly change the plot if the reviewer feels strongly about it. Figure 5 has been removed and instead a new correlation analysis has been performed between the items in the questionnaire and the reaction times and included in a table. 

Reviewer 2 Report

The authors have studied the effect of tactile imagery on the reaction time. The importance of tactile imagery lies in its ability to improve sensory perception, enhance awareness of body sensations, and aid recovery from physical injuries. Its real-life applications can be seen in the fields of psychology, rehabilitation, and sports performance, where it can lead to improved outcomes and better overall performance. Hence, the study has addressed a very important area of neuroscience which can be a promising treatment modality for improving the sensory function and reaction time improving the quality of life of patients or affected population.

Below are the comments/suggestions:

1.       Introduction section is well written and organized. Easy to comprehend and explains the purpose and broader goal of the study. Few grammatical errors and spelling errors noticed. Please revise as you deem fit.

2.       Please re-write the method section to make it clearer. Specially the experimental design section is harder to understand on what actually is going on between the control and the experimental group. Example, Stating that “while subjects in the control group only performed two reaction

time sessions similar to pre- and post-TI….. (line 135)” sounds like control group were also trained. Please revise and make it clear.

3.       Results :

i.         The authors have presented the data for both left and right hand, figure 3A, but the stat has not been performed to compare the data between the left and right hand. Also what are the authors trying to explain showing the data of both hands? it has not been explained / interpreted throughout the manuscript. It has been stated that upon questioning, the subjects reported vividness more in the right hand, but what could be the reason for it? Is it because most of them were right-handed?

ii.       What is the data in Figure 3B trying to suggest? Please explain. Also why is there huge error bar in right hand and not in left hand.

iii.     Figure 2 and Figure has same data except in Figure 4 an evaluation group has been added. Why not just show the data once? Which stat was done for the results in the Figure 4? If stat was done why was there no statistical difference in figure 4 between pre and post group but in figure 2 there was statistically significant?

4.       Discussion section is good except that the effect of TI in different hands is not explained. The limitation has been addressed well and the conclusion section is also good.

Author Response

  1.       Introduction section is well written and organized. Easy to comprehend and explains the purpose and broader goal of the study. Few grammatical errors and spelling errors were noticed. Please revise as you deem fit.

We thank the reviewer for the appreciation. We have corrected the grammatical errors.

  1.       Please re-write the method section to make it clearer. Especially the experimental design section is harder to understand on what actually is going on between the control and the experimental group. Example, Stating that “while subjects in the control group only performed two reaction time sessions similar to pre- and post-TI….. (line 135)” sounds like control group were also trained. Please revise and make it clear.

The methods section have been updated to provide better clarity on the experimental procedure. 

  1.       Results :

  1.         The authors have presented the data for both left and right hand, figure 3A, but the stat has not been performed to compare the data between the left and right hand. Also what are the authors trying to explain showing the data of both hands? it has not been explained / interpreted throughout the manuscript. It has been stated that upon questioning, the subjects reported vividness more in the right hand, but what could be the reason for it? Is it because most of them were right-handed?

We did not perform stats between the two hands since the aim of the study is to evaluate the learning effect of tactile imagery on reaction time. Although stats between the hands would provide additional information, it is quite evident by looking at the values that the chances of a statistical difference is low. However, we appreciate the reviewer’s comment and have instead performed a correlation analysis. We have found that the hand preference and the handedness of the subjects are highly correlated. The information has been updated.

  1.       What is the data in Figure 3B trying to suggest? Please explain. Also why is there a huge error bar in right hand and not in left hand.

Figure has been updated to show the difference between experimental and control group at various time points. 

iii.     Figure 2 and Figure has same data except in Figure 4 an evaluation group has been added. Why not just show the data once? Which stat was done for the results in the Figure 4? If stat was done why was there no statistical difference in figure 4 between pre and post group but in figure 2 there was statistically significant?

Figure 4 shows the data from the 25 subjects out of the 35 subjects from the experimental group when they came for the follow-up. Since this was a subset of the earlier data shown in figure 2, we have included a new figure. The means and SD are different because of the different number of subjects.

  1.       Discussion section is good except that the effect of TI in different hands is not explained. The limitation has been addressed well and the conclusion section is also good.

We thank the reviewer for the words of appreciation. 

Reviewer 3 Report

The authors have done an interesting research, however there are some leaks that I feel they can not be solved.

Following I specify the modifications that are needed to be done, and some comments I hope help the authors to improve their manuscript

Abstract: 

The concept about motor imagery is not correct, motor imagery does not need physical movement, in deed the advantage of this technique is that you do not need to move. Please change the statement.

Introduction:

Line 53.

Motor imagery is not an "emerging technique". It has been used since many many years ago in sports nad dance, however in rehabilitation or therapy is somehow recent, let’s say about 20 years ago we started doing research in this field. See for example articles of Moseley, GL and La Touche, R.

Lines 58-62. 

As I said before one of the advantages of the MI is that the patient do not need to perform the physical movement. We do evaluate the ability to generate mental images and if it is reduced then we need to add action obervation strategies (another movement representation technique). 

I think the intention you have using a novel sensitive strategy instead of a motor strategy is very interesting, but ther is no need to discredit motor strategies such as motor imagery or action observation (the alternative when motor imagery is not a good option). And mostly because this stantement is not true: “to achieve a high degree of functional equivalence with MI, the subjects must be able to perform the action physically (Olsson et al., 2010).” There are strategies to solve this problem as I mentioned with Action observation, and I do not agree with the statement because if you have been able to do a movement in the past you are able to imagine some of it. It is differente in people with birth neurological problems whose never had been able to do specific movements. 

You need to rethink about this points, and change the paragraph.

Line 67. “ This tactile imagery (TI) does not require 66 imagining physical movements or the physical practice required to improve MI abilities”

In rehabilitation the objective is not to improve motor imagery but to improve the movement of the patients throught the training of motor imagery. Actually, one of the indications of MI is for those patients that wear a cast or can not move (for other reasons) but need to train the motor cortex.  It is not correct the justification you say about that motor imagery need a physical training.

Of course the final objective is to improve movement, and to improve movents you need to move. Other way is how you facilite the therapeutic process: with motor cortex stimulation (motor imagery) or sosamtosensory cortex stimulation (tactile imagery). Actually tactile imagery is very close to kinesthesic motor imagery which is the hability of the person to feel the sensations of doing a movement.

Please change the argumets you offer in the introduction regarnding motor imagery and its application.

Explain and justify how could a tactile stimulus have a motor effect without training an actual movement. Is seen there is an icohrence about the therapeutic strategy, because you justify that the patient would not need to move, but the assessment or the final objective is movement improvement? How can this be possible? Do you think that only tactile stimulation without an actual physical movement training would be better than a motor cortical stimulation with a physical movemnte training? What is the point of using only tactile stimulus instead of motor stimulus (thinking about movement representation techniques which include motro imagery and action observation) for the cortex?

 You should change the introduction to include different explanations and justifications. Motor imagery is one interesting tool among the movement representation techniques but different from the tactil or somatosensory training. I would rather explain the importance of the somatosenroy traing for imporving movement and how the implementation of mental trainng atrtagies can lead to improvements, and therefore it would be interesting to prove this tactile imagery as a novel therapeutic strategy.

Attention over a body part may improve movements? Please explain this point. (lines 69-71)

 Couldn’t you be training the stimulus threshold instead of the motor reaction it self??? I mena the imporvemnts could be due to an imporvemnte of the sensory system in terms of learning and imporving the tactile discrimination. Explain more about the effects over learning and sensorymotor system.

Experimental design:

Specify the timing of the measures, how many days elapse between the first assessment, the training session, the second assessment and the follow up assessment.

Specify if the 25 subjects measured in the follow up assessment were from the experimental or the control group. 

Results

It seem you do not compare the experimental grup and control group for each time line point. It is needed to analize the basal reaction time for both groups as weel ans the comaprisson of the post intervention for the experimental group compared with the second assessment results for the control group. You need to include those comparissons, also include the figures for those comparisons.  

The comparison of the RT-e with the pre intervention should have been done for both groups control (first measure) and experimental (first measure) and with the second measure for both groups.

Methodologically, it seems there is a mistake, since the control group should have received a training session without the tactile imagery (maybe a shprter session) because the improvement in the experimental group could only have been due to a learning effect since they had practiced more repetitions of the task. The study does not have a correct design.

The results you show in Figure 5 are not necessary, the interest of colleting those data is to correlate the ability to generate vivid images with the reaction time, please include a correlations analyses. 

Discussion

The first paragraph of the discussion and the conclusions needs to be confirmed after the new statistical analyses have been done (with the correct comparisons with first and second assessments of the control group).

Lines 252-254 

“In the current study, even repeating the reaction time test several times has shown to have a training effect on both the experimental and control groups with both groups showing a reduction in reaction time at the end of the study.” 

This sentence is not explained in the outcomes you offer in the results section. Do you mean there was no difference between groups after all because of a learning effect?

Line 256 

“However, the addition of TI training had a significant effect on the reaction time of the experimental group.”

This sentence is not shown in the results section, regarding the outcomes you show.  There is not a significant difference with a p -value <0.05 for the comparison between the RT-post (experimental group) and the second assessment of the control group. 

Author Response

We thank the reviewers for providing thoughtful comments. We have responded to the comments point-by-point. The corresponding changes have been made in the revised manuscript.

Abstract: 

The concept about motor imagery is not correct, motor imagery does not need physical movement, in deed the advantage of this technique is that you do not need to move. Please change the statement.

The abstract has been updated and the information about motor imagery has been corrected. 

Introduction:

Line 53.

Motor imagery is not an "emerging technique". It has been used since many many years ago in sports nad dance, however in rehabilitation or therapy it is somehow recent, let’s say about 20 years ago we started doing research in this field. See for example articles of Moseley, GL and La Touche, R.

Lines 58-62. 

As I said before one of the advantages of the MI is that the patient do not need to perform the physical movement. We do evaluate the ability to generate mental images and if it is reduced then we need to add action observation strategies (another movement representation technique). 

I think the intention you have using a novel sensitive strategy instead of a motor strategy is very interesting, but there is no need to discredit motor strategies such as motor imagery or action observation (the alternative when motor imagery is not a good option). And mostly because this statement is not true: “to achieve a high degree of functional equivalence with MI, the subjects must be able to perform the action physically (Olsson et al., 2010).” There are strategies to solve this problem as I mentioned with Action observation, and I do not agree with the statement because if you have been able to do a movement in the past you are able to imagine some of it. It is different in people with birth neurological problems whose never had been able to do specific movements. 

You need to rethink about this points, and change the paragraph.

Line 67. “ This tactile imagery (TI) does not require 66 imagining physical movements or the physical practice required to improve MI abilities”

In rehabilitation the objective is not to improve motor imagery but to improve the movement of the patients throughout the training of motor imagery. Actually, one of the indications of MI is for those patients that wear a cast or can not move (for other reasons) but need to train the motor cortex.  It is not correct the justification you say about that motor imagery need a physical training.

Of course the final objective is to improve movement, and to improve movements you need to move. Other way is how you facilitate the therapeutic process: with motor cortex stimulation (motor imagery) or somatosensory cortex stimulation (tactile imagery). Actually tactile imagery is very close to kinesthetic motor imagery which is the ability of the person to feel the sensations of doing a movement.

Please change the arguments you offer in the introduction regarding motor imagery and its application.

Explain and justify how could a tactile stimulus have a motor effect without training an actual movement. Is seen there is an incoherence about the therapeutic strategy, because you justify that the patient would not need to move, but the assessment or the final objective is movement improvement? How can this be possible? Do you think that only tactile stimulation without an actual physical movement training would be better than a motor cortical stimulation with a physical movement training? What is the point of using only tactile stimulus instead of motor stimulus (thinking about movement representation techniques which include motro imagery and action observation) for the cortex?

You should change the introduction to include different explanations and justifications. Motor imagery is one interesting tool among the movement representation techniques but different from the tactile or somatosensory training. I would rather explain the importance of the somatosensory training for improving movement and how the implementation of mental training strategies can lead to improvements, and therefore it would be interesting to prove this tactile imagery as a novel therapeutic strategy.

Attention over a body part may improve movements? Please explain this point. (lines 69-71)

Couldn’t you be training the stimulus threshold instead of the motor reaction itself??? I mean the improvements could be due to an improvement of the sensory system in terms of learning and improving the tactile discrimination. Explain more about the effects over learning and sensorimotor system.

We appreciate the reviewer for the extensive suggestions to improve the introduction. We realize that we have unintentionally demerited the effects of motor imagery and have rewritten the entire introduction to correct this. We have taken the reviewer’s suggestions and have updated the information about motor imagery and have framed tactile imagery as a novel rehabilitation paradigm that focuses on sensory learning while motor imagery focuses on motor learning, making the two compliment each other. We hope the updated introduction satisfies the reviewer.

Experimental design:

Specify the timing of the measures, how many days elapse between the first assessment, the training session, the second assessment and the follow up assessment.

All the initial assessments including reaction time measurements and questionnaire were performed on the same day and the follow-up assessment was performed 4 weeks later. We have updated the information in the methods section (lines 169-179).

Specify if the 25 subjects measured in the follow up assessment were from the experimental or the control group. 

They are from the experimental group and the information is presented in the methods sections line 170. 

Results

It seem you do not compare the experimental group and control group for each time line point. It is needed to analyze the basal reaction time for both groups as well as the comparison of the post intervention for the experimental group compared with the second assessment results for the control group. You need to include those comparisons, also include the figures for those comparisons.  

We have updated our statistical analyses to compare between the groups at each time point and have included a figure 3.

The comparison of the RT-e with the pre-intervention should have been done for both groups control (first measure) and experimental (first measure) and with the second measure for both groups.

We appreciate the reviewer’s suggestion. However, we were aiming to evaluate the training effect of tactile imagery and so the follow-up was performed only with the experimental group to see if the improved reaction time (compared to control) also persisted over time. RT-e is only from the 25 subjects from the experimental group who came back for the follow-up and so it was compared to the initial reaction times for those 25 subjects alone. 

Methodologically, it seems there is a mistake, since the control group should have received a training session without the tactile imagery (maybe a shorter session) because the improvement in the experimental group could only have been due to a learning effect since they had practiced more repetitions of the task. The study does not have a correct design.

We understand the importance of the study design and would like to take this opportunity to explain our design. The aim of the study was to evaluate the training effect of tactile imagery on reaction time. We believed that a proper study design to achieve our aim was to have two groups (experimental vs. control) where one goes through tactile imagery sessions and the other doesn’t. However, as previous studies have shown and from what we have seen in our study too, the sessions where we evaluate the baseline and post tactile imagery session reaction times itself have a learning effect. Hence to account for that, we included an additional time point for the control group to correspond with the post-TI reaction time evaluation for the experimental group. Hence, we made sure that both the groups have the same number of repetition of the reaction time tasks eliminating any confounding learning effects. This was done so that any changes in the reaction time (reduction in reaction time) in the experimental group compared to the control group in the post-tactile imagery training could be attributed to the training effect of tactile imagery. We would also like to clarify that during the tactile imagery training, the subjects in the experimental group do not perform any reaction time tasks. The subjects are instructed to only imagine a vibratory stimulus on their index fingers. We have updated the methods section to reflect this information to offer better clarity. 

The results you show in Figure 5 are not necessary, the interest of collecting those data is to correlate the ability to generate vivid images with the reaction time, please include a correlations analyses. 

We have performed the correlation analysis and have replaced the figure with a table with the correlation results. 

Discussion

The first paragraph of the discussion and the conclusions needs to be confirmed after the new statistical analyses have been done (with the correct comparisons with first and second assessments of the control group).

Lines 252-254 

The information has been updated. 

“In the current study, even repeating the reaction time test several times has shown to have a training effect on both the experimental and control groups with both groups showing a reduction in reaction time at the end of the study.” 

This sentence is not explained in the outcomes you offer in the results section. Do you mean there was no difference between groups after all because of a learning effect?

We have rewritten the sentence and have added additional results to support this statement. We have compared the baseline reaction times between the experimental and control groups and found no statistical difference. We also compared the reaction times post TI training in the experimental group and the second round of reaction time evaluation blocks from the control group and found a statistical difference. 

Line 256 

“However, the addition of TI training had a significant effect on the reaction time of the experimental group.”

This sentence is not shown in the results section, regarding the outcomes you show.  There is not a significant difference with a p -value <0.05 for the comparison between the RT-post (experimental group) and the second assessment of the control group. 

The results have been rewritten and new information has been added to support this statement. 

Round 2

Reviewer 1 Report

No further comments

Reviewer 3 Report

Thanks for attending all the queries.